# The Impact of Omega-3 Fatty Acids on the Evolution of *Acinetobacter baumannii* Drug Resistance

Maoge Zang,[a] Felise G. Adams,[a] Karl A. Hassan,[b] Bart A. Eijkelkamp[a]

[a]Molecular Sciences and Technology, College of Science and Engineering, Flinders University, Adelaide, South Australia, Australia
[b]School of Environmental and Life Sciences, University of Newcastle, Callaghan, New South Wales, Australia

**ABSTRACT** The bacterial pathogen *Acinetobacter baumannii* has emerged as an urgent threat to health care systems. The prevalence of multidrug resistance in this critical human pathogen is closely associated with difficulties in its eradication from the hospital environment and its recalcitrance to treatment during infection. The development of resistance in *A. baumannii* is in part due to substantial plasticity of its genome, facilitating spontaneous genomic evolution. Many studies have investigated selective pressures imposed by antibiotics on genomic evolution, but the influence of high-abundance bioactive molecules at the host-pathogen interface on mutation and rates of evolution is poorly understood. Here, we studied the roles of host fatty acids in the gain in resistance to common antibiotics. We defined the impact of the polyunsaturated fatty acids arachidonic acid and docosahexaenoic acid on the development of resistance to erythromycin in *A. baumannii* strain AB5075_UW using a microevolutionary approach. We employed whole-genome sequencing and various phenotypic analyses to characterize microbe-lipid-antibiotic interactions. Cells exposed to erythromycin in the presence of the fatty acids displayed significantly lower rates of development of resistance to erythromycin and, importantly, tetracycline. Subsequent analyses defined diverse means by which host fatty acids influence the mutation rates. This work has highlighted the critical need to consider the roles of host fatty acids in *A. baumannii* physiology and antimicrobial resistance. Collectively, we have identified a novel means to curb the development of resistance in this critical human pathogen.

**IMPORTANCE** The global distribution of multidrug resistance in *A. baumannii* has necessitated seeking not only alternative therapeutic approaches but also the means to limit the development of resistance in clinical settings. Highly abundant host bioactive compounds, such as polyunsaturated fatty acids, are readily acquired by *A. baumannii* during infection and have been illustrated to impact the bacterium's membrane composition and antibiotic resistance. In this work, we show that *in vitro* supplementation with host polyunsaturated fatty acids reduces the rate at which *A. baumannii* gains resistance to erythromycin and tetracycline. Furthermore, we discover that the impact on resistance development is closely associated with the primary antimicrobial efflux systems of *A. baumannii*, which represent one of the major drivers of clinical resistance. Overall, this study emphasizes the potential of host macromolecules in novel approaches to circumvent the difficulties of multidrug resistance during *A. baumannii* treatment, with fatty acid supplements such as fish oil providing safe and cost-effective ways to enhance host tolerance to bacterial infections.

**KEYWORDS** antimicrobial host lipids, free fatty acids, macrolides, resistance evolution, RND efflux, AdeABC, AdeIJK

Address correspondence to Bart A. Eijkelkamp, bart.eijkelkamp@flinders.edu.au.

*A*cinetobacter baumannii is a Gram-negative, opportunistic pathogen that prevails in nosocomial settings and causes severe infections among patients admitted to the intensive care unit (1). *A. baumannii* came to notoriety as a troublesome wound-

infecting pathogen in soldiers serving in Iraq (2), was subsequently included as an ESKAPE pathogen along with other major opportunistic pathogens (3), and has now been recognized as a critical pathogen by the WHO and CDC (4, 5). Infections by *A. baumannii* can lead to meningitis, pneumonia, and sepsis, with alarming mortality rates because victims are commonly immunocompromised (6) or suffer from comorbidities that exacerbate disease symptoms (7). The global distribution of multidrug-resistant (MDR) *A. baumannii* is alarming, as is its rapid development of clinical resistance. This can be attributed to the pathogen's high levels of intrinsic resistance, robust ability to acquire foreign resistance determinants, and genome plasticity, allowing mutations to lead to resistance (8, 9). Proteins from the resistance-nodulation-cell division (RND) superfamily, known typically as Ade efflux systems in *Acinetobacter* species, play key roles in the development of MDR (10–13). The three primary Ade multidrug efflux systems in *A. baumannii* are AdeABC, AdeFGH, and AdeIJK. These pumps also play key roles in bacterial virulence and physiology (12). The development of resistance in *A. baumannii* is driven largely by its existence in the hospital environment, where the use of a diverse array of antimicrobial compounds has been associated with constitutive upregulation of genes encoding RND efflux systems, particularly *adeABC* and *adeIJK*, via regulatory mutations (11, 14, 15). Specifically, selective pressure exerted by antimicrobials selects for direct mutations within the efflux systems or their corresponding regulatory components, including *adeRS* (*adeABC*) and *adeN* (*adeIJK*) (11, 14, 16, 17).

Antimicrobial resistance is mostly studied in standard laboratory media, and the impact of host molecules is thus overlooked. Fatty acids are highly abundant in most host niches, they are nearly indiscriminately acquired by *A. baumannii*, and they can impact *A. baumannii* physiology (18–21). The nearly universal antimicrobial activity of the long-chain polyunsaturated fatty acids (PUFAs) (with ≥16 carbons and ≥2 double bonds) that belong to the omega-3 and omega-6 groups has been well established (19, 22). The concentration of PUFAs in the human plasma can vary considerably, from 0.1 mM to 0.6 mM, depending on dietary intake, with the shift toward western diets rapidly decreasing their accumulation within the host (23, 24). The presence of PUFAs is unfavorable to the bacteria, due to their high levels of structural dissimilarity in comparison with native fatty acid species, which are comparatively shorter and more saturated (25). Consistently, despite accumulating to lower abundance, docosahexaenoic acid (DHA) (with 22 carbons and 6 double bonds [omega-3]) demonstrates greater antimicrobial activity than arachidonic acid (AA) (with 20 carbons and 4 double bonds [omega-6]) with *A. baumannii* (19). The RND efflux system AdeIJK has been found to contribute to phospholipid homeostasis, which provides indirect protection against DHA stress (19). In contrast, recent analyses have shown that DHA incorporation in the *A. baumannii* phospholipid membrane affects the structure and function of the AdeABC RND efflux system (18). Considering the ability of *A. baumannii* to adapt rapidly to clinical treatment during infection and the intricate interplay between lipid homeostasis and antimicrobial resistance, such as that mediated by RND pumps, we sought to determine the impact of PUFAs on the ability of *A. baumannii* to gain resistance.

## RESULTS

**PUFAs impact antibiotic resistance and laboratory-based microevolution of MDR.** To establish the relative impact of distinct host-derived fatty acids on the antibiotic resistance profile of *A. baumannii*, we examined the MICs following AA (omega-6) or DHA (omega-3) supplementation. Both PUFAs resulted in 2- to 8-fold increased susceptibility to aminoglycoside antibiotics (gentamicin and streptomycin) (Table 1). Changes in resistance to macrolide antibiotics (erythromycin and azithromycin) were minimal, with only a 2-fold reduction in azithromycin resistance observed in the presence of AA. PUFA supplementation did not affect resistance to tetracycline, and only minimal changes in chloramphenicol resistance were observed (2-fold changes following DHA supplementation). Interestingly, supplementation with PUFAs negatively affected the efficacy of colistin (4-fold change with AA and 8-fold change with DHA). These resistance studies show that the omega-6 PUFA AA impacts *A. baumannii*

**TABLE 1** MIC analyses of *A. baumannii* AB5075_UW with or without PUFAs

| Drug | MIC ($\mu$g · ml$^{-1}$) | | |
|---|---|---|---|
| | Untreated | Treated with 0.25 mM AA | Treated with 0.25 mM DHA |
| Erythromycin | 32–64 | 32 | 32[a] |
| Azithromycin | 64[a] | 32 | 64[a] |
| Colistin | 2 | 16 | 8 |
| Streptomycin | 512[a] | 256 | 128[a] |
| Gentamicin | 1,024[a] | 128 | 128[a] |
| Chloramphenicol | 128[a] | 128 | 64[a] |
| Tetracycline | 0.3125[a] | 0.3125 | 0.3125[a] |

[a]Data from previous work (18).

resistance similarly to the omega-3 PUFA DHA, despite the differences in their accumulation and antimicrobial activities (19).

We next sought to study the potential of PUFAs to influence the development of antibiotic resistance in *A. baumannii*. We selected erythromycin as the screening compound, because its efficacy was unaffected by either PUFA, thus allowing the use of a consistent concentration across all conditions. Growth dynamic analyses revealed that 8 $\mu$g · ml$^{-1}$ erythromycin induced a minimal but noticeable degree of perturbation (see Fig. S1 in the supplemental material). Overnight culturing on solid medium containing erythromycin (8 $\mu$g · ml$^{-1}$) resulted in more than 50% of the population gaining an increase in resistance to the antibiotic (Fig. 1A). This was defined by scoring the growth on medium containing erythromycin at 24 $\mu$g · ml$^{-1}$, a concentration that is nonpermissive to growth of untreated AB5075_UW cells (Fig. 1A). Although the relative

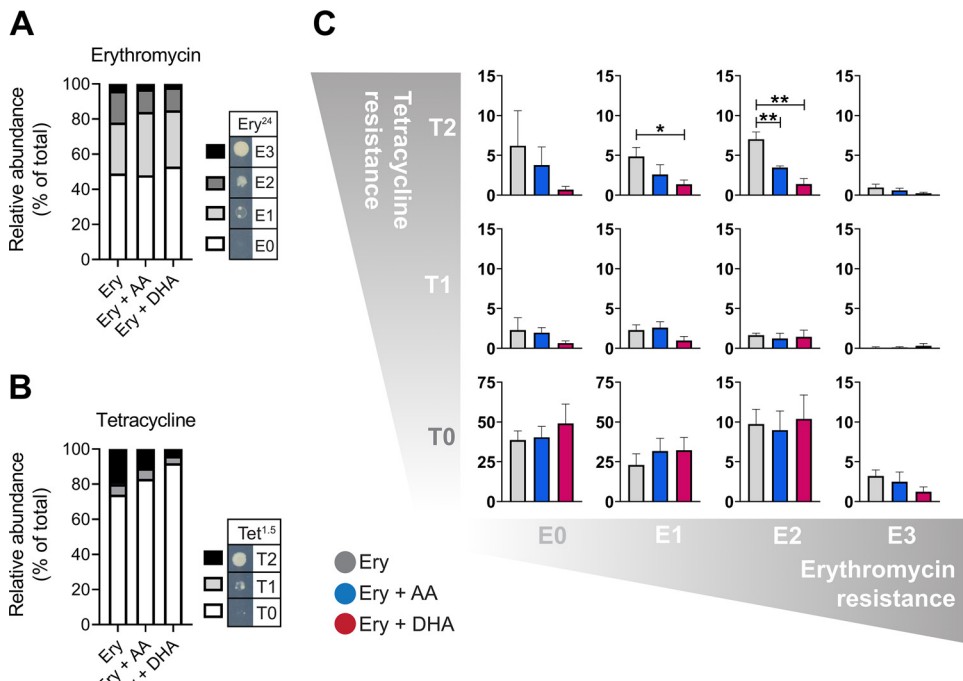

**FIG 1** Microevolution of antibiotic resistance in *A. baumannii*. (A and B) The gain of resistance to erythromycin (Ery) at 24 $\mu$g · ml$^{-1}$ (A) and tetracycline at 1.5 $\mu$g · ml$^{-1}$ (B) in *A. baumannii* strain AB5075_UW was assessed after preexposure to erythromycin (8 $\mu$g · ml$^{-1}$) overnight, with or without supplementation with 250 $\mu$M AA or DHA. Individual clones were scored based on their relative growth intensities in spot tests (erythromycin: E0, no growth; E1, minor growth; E2, intermediate growth; E3, full growth; tetracycline: T0, no growth; T1, intermediate growth; T3, full growth). (C) Subpopulations of clones were defined based on the combined growth intensity scores for erythromycin and tetracycline for each sample group (erythromycin only, erythromycin plus AA, and erythromycin plus DHA). Data represent the mean of 4 independent colony libraries for each condition. Statistical analyses were carried out using two-tailed Student's *t* tests. *, *P* < 0.05; **, *P* < 0.01.

abundance of clones with increased erythromycin resistance appeared to decrease when cells were grown in combination with either AA or DHA, this effect was only minor (Fig. 1A). Interestingly, exposure of *A. baumannii* AB5075_UW to 8 $\mu$g · ml$^{-1}$ erythromycin also resulted in increased abundance of tetracycline-resistant clones (>25% of the total population) (Fig. 1B). PUFA-erythromycin cotreatment displayed profound impacts on the development of tetracycline resistance (Fig. 1B). This suggests that the presence of PUFAs, particularly DHA, may impact the prevalence of clones with enhanced resistance to erythromycin and tetracycline. To delineate subpopulations of clones based on different erythromycin and tetracycline susceptibilities (12 subpopulations in total), the results were aligned for each independent clone isolated from the three distinct treatment groups (Fig. 1C). The most dramatic changes following cotreatment with AA or DHA were identified in a population of clones with elevated erythromycin and tetracycline resistance (i.e., E2T2) (Fig. 1C). The average from four independently conducted experiments found that the percentages of clones in this subpopulation decreased 2-fold or 5-fold following cotreatment with AA or DHA, respectively, compared to erythromycin treatment alone (Fig. 1C). MIC analysis of 23 randomly selected E2T2 clones revealed average increases of 2-fold for erythromycin and 3-fold for tetracycline, compared to the E0T0 control group (22 clones) (see Fig. S2). In addition to erythromycin and tetracycline, the E2T2 clones displayed an increase in resistance to azithromycin.

**RND efflux pump overexpression is key to resistance gain.** To identify the genetic basis of the observed increase in resistance to erythromycin and tetracycline, whole-genome sequencing was performed on 6 randomly selected clones from the E2T2 subpopulation and 1 randomly chosen clone from the E0T0 subpopulation. Comparative genome analysis of the E2T2 subpopulation clones versus the AB5075_UW reference revealed that 4 of the 6 sequenced clones harbored a unique nonsynonymous single-nucleotide polymorphism (SNP) that mapped to *adeRS*, i.e., genes that encode the AdeRS two-component regulatory system (AdeS$_{P154S}$, AdeS$_{D167N}$, AdeS$_{F170L}$, and AdeR$_{I27S}$) (Fig. 2A). The remaining E2T2 strains harbored insertion sequence (IS)-mediated mutations associated with the *adeIJK/adeN* regulon. Genome analysis revealed integration of resident IS*Aba1* and IS*Aba13* elements upstream of the membrane fusion protein gene *adeI* (*adeI*::IS*Aba1*) and the TetR-type transcriptional regulator gene *adeN* (*adeN*::IS*Aba13*), respectively (Fig. 2B and C). The integrated IS*Aba1* element upstream of *adeI* was found to be in an appropriate orientation for the outward-facing promoter present in IS*Aba1* to increase transcription of the *adeIJK* RND efflux operon (Fig. 2D). IS*Aba13* integration occurred between the putative native promoter sequence responsible for transcription initiation of *adeN* and the corresponding start codon (Fig. 2C and E). Transcriptional profiling revealed that IS*Aba13* integration reduced *adeN* expression 13.4-fold, compared to that in the parental AB5075_UW cells (see Fig. S3).

To ascertain the impact the identified mutations had on the *adeABC* and *adeIJK* RND efflux systems, transcriptional profiling of *adeB* and *adeJ* was performed on the panel of 6 E2T2 clones. Expression of *adeB* was found to be significantly upregulated (55- to 155-fold) in all 4 strains harboring distinct substitutions in AdeR/S, with AdeS$_{D167N}$ exhibiting the greatest change (Fig. 3A). Expression of *adeJ* was found to be significantly upregulated in the *adeN*::IS*Aba13* and *adeI*::IS*Aba1* clones, with IS*Aba1* integration having the greatest impact (38-fold). IS*Aba13*-mediated downregulation of *adeN* expression resulted in a 5-fold increase in *adeJ* expression (Fig. 3B). To gain a greater understanding of the relative contribution of each unique mutation to antibiotic susceptibility, growth analyses in the presence of tetracycline (1 $\mu$g · ml$^{-1}$) were performed. All strains displayed marked increases in tetracycline resistance, with AdeR$_{I27S}$ and *adeI*::IS*Aba1* mutants showing the most significant increases (Fig. 3C). These findings suggest that both AdeABC and AdeIJK have efficient tetracycline efflux capacity, provided that their genes are transcriptionally activated.

***A. baumannii* RND efflux systems are closely linked to cellular lipid homeostasis.** To examine the mechanism by which PUFAs reduce the development of RND-

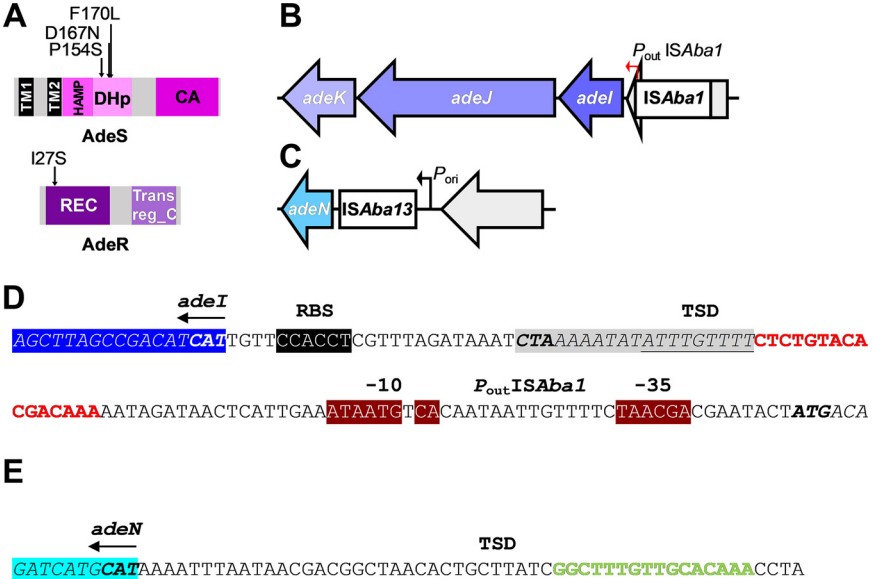

**FIG 2** Genetic features of antibiotic-resistant clones. (A) Mapping of nonsynonymous mutations identified in the AdeRS two-component signal transduction system from sequenced E2T2 strains. Protein domain architectures of AdeR and AdeS are based on output from the SMART database (40) using sequences obtained from AB5075_UW annotation (GenBank accession number CP008706). The mutation changes and relative positions within AdeS and AdeR proteins are shown. The AdeS$_{P154S}$ substitution is located within the H-box motif of AdeS and is a highly conserved residue across classic HisKA-type DHp domains (pfam00512) (41). Both AdeS$_{D167N}$ and AdeS$_{F170L}$ substitutions are located at the C-terminal end of the DHp domain of AdeS. The AdeR$_{I27S}$ substitution is located within the REC domain of AdeR, 6 residues downstream from the first of three conserved D-box motifs that generate an acidic triad essential for phosphorylation. (B and C) Illustrations of the positions of the integrated IS elements, i.e., IS*Aba1* (B) and IS*Aba13* (C), and the surrounding genetic loci identified from sequenced E2T2 strains. Open arrows depict neighboring genes of interest and the direction of transcription. IS elements are displayed as rectangular boxes. Positions and orientations of the outward-facing promoter identified in IS*Aba1* (P$_{out}$ IS*Aba1*) (B) and the native *adeN* promoter (P$_{ori}$) (C), which are represented by closed red and black arrows, respectively, in panels B and C. Relevant gene names are labeled. (D) An extended −10 promoter sequence present in some IS*Aba1* elements, which initiates high-level transcription of neighboring upstream genes (35, 42), was identified within the translocated IS*Aba1* genetic sequence. (D and E) The transposon target site duplication (TSD) sequences are underlined. Gene sizes and intergenic distances are not drawn to scale. TM, transmembrane domain; HAMP, histidine kinase, adenylyl cyclase, methyl-accepting protein, and phosphatase; CA, catalytic domain; REC, receiver domain; Trans reg_C, DNA-binding domain; RBS, ribosome binding site.

overexpressing clones, growth of AB5075_UW cells, an E0T0 control strain, and the 6 sequenced E2T2 clones was examined in the presence and absence of DHA (Fig. 4A). These analyses identified strain *A. baumannii* AdeR$_{I27S}$ as being hypersusceptible to DHA, while the other E2T2 clones displayed growth perturbations similar to those of the AB5075_UW or E0T0 control strains. This indicates that at least a proportion of mutants that overexpress *adeABC* are unlikely to propagate when exposed to erythromycin and DHA.

We previously identified a role for AdeIJK in *A. baumannii* DHA resistance (19) and hypothesized that its transcriptional derepression in the presence of PUFAs provides *A. baumannii* with transiently enhanced erythromycin resistance. Indeed, the transcriptional responses of *A. baumannii* AB5075_UW to either 8 $\mu$g · ml$^{-1}$ erythromycin, 250 $\mu$M DHA, or the combination thereof revealed that erythromycin alone does not result in the transcriptional activation of *adeJ* but DHA leads to substantial derepression (Fig. 4B). This indicates that PUFAs may reduce the selective pressure for transposon-mediated *adeIJK* upregulation to occur when cells are exposed to erythromycin.

## DISCUSSION

Microbial exposure to macrolides can occur as a direct result of their use as antimicrobials but also as a secondary effect of their use as immune modulators or regulators

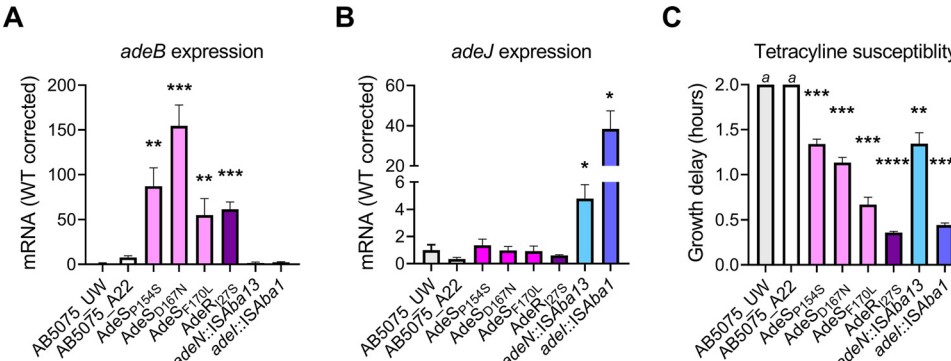

**FIG 3** Overexpression of *adeB* and *adeJ* in resistant clones leads to tetracycline resistance. (A and B) The mRNA expression levels of *adeB* (A) and *adeJ* (B) were examined using qRT-PCR with mid-log-phase cultures of the wild-type strain (AB5075_UW), 6 sequenced representative E2T2 clones (AdeS$_{P154S}$, AdeS$_{D167N}$, AdeS$_{F170L}$, AdeR$_{I27S}$, *adeN*::IS*Aba13*, and *adeI*::IS*Aba1*), and an E0T0 control (AB5075_A22). All data are the mean of at least biological triplicates ± the standard error of the mean (SEM). Statistical analyses were performed using Student's *t* test. *, $P < 0.05$; **, $P < 0.01$; ***, $P < 0.001$. (C) Examination of tetracycline susceptibility was performed on the same panel of strains/clones. The tetracycline-induced delay was calculated by defining the time to reach 50% of the maximum growth in untreated cultures and cultures exposed to 1.5 $\mu$g · ml$^{-1}$ tetracycline in LB broth using a microplate growth assay (OD$_{600}$ measured every 30 min). The *a* denotes growth delays greater than 2 h for the AB5075_UW and AB5075_A22 control strains. All data are the mean of biological triplicates ± SEM. Statistical analyses were performed using one-way analysis of variance (ANOVA). **, $P < 0.01$; ***, $P < 0.001$; ****, $P < 0.0001$.

of pulmonary surfactant homeostasis (26, 27). The latter two uses are related to host lipid metabolism, i.e., preventing AA release for subsequent cyclooxygenase activation (28) or changing the lipid composition of pulmonary surfactant by as yet unknown mechanisms, respectively. As a result, macrolides are of significant interest for the treatment of cystic fibrosis patients (29). Despite their known interactions with lipids, our work showed that macrolide antibiotics retain similar anti-*Acinetobacter* activity in the presence of PUFAs. This may be a result of these molecules being substrates of both AdeABC and AdeIJK (13, 30); although AdeABC efflux activity is reduced in the presence of PUFA (18), the transcriptional activation of *adeIJK*, as illustrated in this study, may counteract this. In contrast to the minimal impact on resistance to macrolides, we showed that antimicrobial host fatty acids of both the omega-6 and omega-3 classes improved the efficacy of aminoglycosides in *A. baumannii*. This suggests that omega-6 PUFAs affect the structure and function of AdeABC, similar to findings

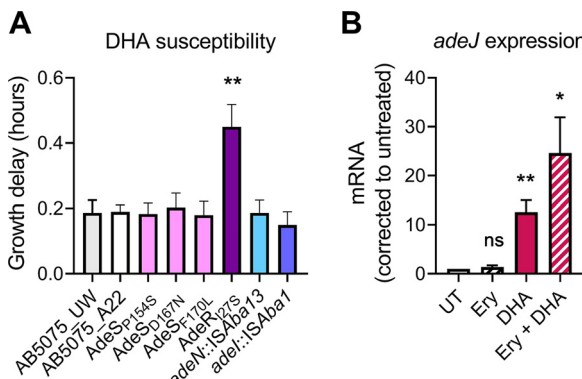

**FIG 4** DHA limits the development of RND-mediated resistance. (A) Three-hour growth analyses were performed with the wild-type strain (AB5075_UW), 6 sequenced representative E2T2 clones (AdeS$_{P154S}$, AdeS$_{D167N}$, AdeS$_{F170L}$, AdeR$_{I27S}$, *adeN*::IS*Aba13*, and *adeI*::IS*Aba1*), and an E0T0 control (AB5075_A22), with or without DHA supplementation (250 $\mu$M). (B) The *adeJ* mRNA expression levels in the wild-type (AB5075_UW) cells were examined following 30-min incubations with erythromycin (Ery) (8 $\mu$g · ml$^{-1}$) and/or DHA (250 $\mu$M). UT, untreated. All data are the mean of biological triplicates ± SEM. Statistical analyses were performed using Student's *t* test. ns, not significant; *, $P < 0.05$; **, $P < 0.01$.

described for DHA (18). Importantly, both PUFAs alleviated the antimicrobial activity of colistin, possibly through extracellular sequestration, which renders these cationic antimicrobial peptides too hydrophobic to readily pass the hydrophilic capsule and lipooligosaccharide barriers of the bacterium. This observation displays similarity to the enhanced resistance of *Streptococcus pneumoniae* to the antimicrobial peptide LL-37 following AA treatment (31). Considering the profound impact of fatty acids on antibiotic activity and their ubiquitous presence at the host-pathogen interface, compared to standard laboratory media (19), the supplementation of fatty acids based on niche-specific profiles (21) in *in vitro* resistance assays should be considered. Exactly how host fatty acids impact macrolide-*A. baumannii* interactions during infection remains unknown, but the work presented here underscores the importance of administering the appropriate antibiotic during treatment, because failed erythromycin treatment may compromise the efficacy of treatment with other antibiotics, such as tetracycline.

Despite a minimal impact on the immediate efficacy of macrolides, in this study we reveal novel mechanisms by which PUFAs affect *A. baumannii* in its ability to gain resistance to erythromycin and the unrelated antibiotic tetracycline. In single-isolate cultures, intrinsic resistance was mediated by adaptive mutations leading to either *adeABC* or *adeIJK* overexpression. The insertional disruption of the *adeN* locus, which encodes the transcriptional repressor of AdeIJK, by ISs is a known means by which clinical and laboratory strains can gain resistance (32–35). However, the introduction of IS*Aba1* upstream of *adeI* has not been described previously, which may indicate that the dramatic *adeIJK* expression influences the virulence of this strain.

To our knowledge, the >10-fold upregulation of *adeJ* following DHA treatment is more dramatic than that observed for any antimicrobial compound studied to date, including AdeIJK efflux substrates, which does not include DHA itself (19). This indicated that AdeN ligands may include fatty acids and/or other hydrophobic compounds, and it sheds light on the putative endogenous/physiological function of AdeN in lipid homeostasis, which may also involve its cotranscribed phosphatidylglycerol-phophate phosphatase (*pgpB*). This observation appears to be linked to the reduction in the mutation rate, as we speculate that the presence of PUFAs minimizes the need for adaptive evolution to increase *adeIJK* expression when cells are exposed to erythromycin. Furthermore, this highlights the potential risk of introducing resistance with PUFA treatment via *adeIJK* upregulation, which would require adequate selection of antimicrobial compounds that may bypass this particular efflux system.

Our analyses identified various resistant clones with a mutation in the AdeRS two-component system. This system is a well-known hot spot for mutations responsible for the upregulation of *adeABC*. The histidine kinase AdeS exists as a homodimer, and two of our mutations were identified in the dimerization and histidine phosphotransfer (DHp) domain, which was associated previously with enhanced AdeS activation (36). In addition, we identified a SNP within the response regulator AdeR at an uncharacterized residue (I27) that is not in close proximity to the DNA-binding domain (residues 138 to 247). Considering that D63 represents the phosphorylation site of AdeR, the AdeR$_{I27S}$ substitution in our study is likely to mediate a more direct impact on phosphorylation.

Subsequent examination showed that the AdeR$_{I27S}$ mutation leads to increased DHA susceptibility, which could, in part, suggest that the upregulation of *adeABC* in response to antibiotic stress is unfavorable in the presence of host lipids. Our previous analysis indicated that the incorporation of DHA in the *A. baumannii* membrane has a profound impact on the structural integrity of AdeB, leading to impaired efflux activity and increased susceptibility to aminoglycosides (18). This adverse effect of DHA on AdeABC could be associated with a reduction in *adeRS* mutations, since there is no benefit to upregulating a functionally disrupted efflux system. Overall, this observation was in contrast to the role of AdeIJK in providing *A. baumannii* with increased tolerance to DHA and its structural and functional resistance to a membrane enriched with DHA (19). The potentially opposing roles of AdeABC and AdeIJK in *A. baumannii* membrane biology were also observed in a phospholipidomic analysis following their

independent overexpression in a triple-RND-mutant background (13). In particular, the abundance of the neutral phosphatidylethanolamine species decreased following overexpression of AdeIJK, whereas the abundance increased following overexpression of AdeABC.

Collectively, our study has revealed that host fatty acids have an impact on antibiotic efficacy and the development of resistance. Hence, this work provides new insights into the potential of nutritional supplementation in combatting the critical human pathogen *A. baumannii*. Fundamentally, this work has increased our understanding of the interplay between antimicrobial stress adaptation, RND efflux, and lipid homeostasis.

## MATERIALS AND METHODS

**Strains, culture media, and generation of resistant clones.** *A. baumannii* AB5075_UW and mutant derivatives (see Table S1 in the supplemental material) were cultured in Lennox medium (LB medium) or LB agar (1.5%). A library of AB5075_UW clones was generated by transferring colonies grown overnight on LB agar with erythromycin (8 $\mu$g $\cdot$ ml$^{-1}$), with or without 250 $\mu$M AA or DHA, to a 96-well plate for subsequent nonselective propagation (see Fig. S4). Screening of the colony library for erythromycin and tetracycline resistance was achieved by spot plating (approximately 8 $\times$ 10$^5$ CFU per spot) onto solid medium. The concentrations selected, i.e., 24 $\mu$g $\cdot$ ml$^{-1}$ erythromycin and 1.5 $\mu$g $\cdot$ ml$^{-1}$ tetracycline, did not support wild-type AB5075_UW growth.

**Bacterial growth assays.** Cultures in LB medium were incubated at 37°C with shaking in a FLUOstar Omega spectrophotometer (BMG Labtech), and mean optical density at 600 nm (OD$_{600}$) values were determined every 30 min. The growth delay was determined by calculating the difference in the time for cultures to reach 50% of the maximal growth under treated versus untreated conditions using the 50% inhibitory concentration (IC$_{50}$) calculation function in Prism v8.4.1 (GraphPad), as described previously (19, 37). The 20-ml cultures used for all other analyses were incubated at 37°C in an Innova 40R shaking incubator (Eppendorf) at 230 rpm until they reached mid-log phase (OD$_{600}$ of 0.7).

**MIC analysis.** The antibiotic resistance profile of *A. baumannii* AB5075_UW was determined using the microdilution method with cation-adjusted Mueller-Hinton (MH) medium, as described previously (38). The impact of PUFAs was assessed by supplementing MH medium with AA or DHA at 250 $\mu$M. The plates were sealed with a breathable film, placed in a humidity box, and incubated overnight at 37°C. MIC values were determined by visual examination.

**Transcriptional analyses.** Overnight *A. baumannii* AB5075_UW and resistant clone cultures were grown to mid-log phase (OD$_{600}$ of 0.7) in 20-ml LB cultures prior to RNA extraction. To examine the impact of erythromycin and DHA, AB5075_UW cultures were grown to an OD$_{600}$ of 0.5 in 20-ml LB cultures, which were split into three 4-ml cultures; one culture was treated with 8 $\mu$g $\cdot$ ml$^{-1}$ erythromycin, one culture was treated with 250 $\mu$M DHA, and the last culture was treated with both compounds concurrently. Cells were grown for another 30 min before RNA was extracted.

Isolation of RNA was performed using previously described methods (20). Briefly, cell pellets were lysed with QiaZol (Qiagen). Following phase separation, RNA was extracted from the aqueous phase using the RNeasy purification minikit (Qiagen), incorporating the on-column RNase-free DNase I (Qiagen) treatment according to the manufacturer's recommendations. Quantitative reverse transcription PCR (qRT-PCR) was performed on a QuantStudio 7 Flex system (Thermo Fisher Scientific) with the Superscript III Platinum SYBR One-Step qRT-PCR kit (Thermo Fisher Scientific). Gene expression was normalized using the constitutively expressed housekeeping gene *GAPDH*. Oligonucleotides for qRT-PCR are listed in Table S2 in the supplemental material.

**DNA extraction and whole-genome sequencing.** DNA was extracted on a QIAsymphony SP system with the QIAsymphony DSP virus/pathogen kit (Qiagen) according to the manufacturer's instructions. The DNA concentration was quantified using the Quant-IT double-stranded DNA (dsDNA) high-sensitivity kit (Thermo Fisher Scientific).

Sequencing libraries were prepared from the pure *A. baumannii* DNA extract samples using the Nextera XT library preparation kit (Illumina Inc.) with slight modifications. One-half of the volume was used for tagmentation reagents, amplification reagents, and input DNA. Library cleanup was performed using the AxyPrep MAG PCR cleanup kit (Corning Inc., NY, USA), and libraries were pooled manually and sequenced on a NextSeq 550 platform with the NextSeq 500/550 midoutput kit v2.5 (300 cycles) (Illumina Inc.). Genome mutations were identified using breseq (39).

## SUPPLEMENTAL MATERIAL

Supplemental material is available online only.
**SUPPLEMENTAL FILE 1**, PDF file, 0.5 MB.

## ACKNOWLEDGMENTS

We are grateful to Lex Leong (SA Pathology) for assistance with whole-genome sequencing.

This work was supported by the National Health and Medical Research Council (Australia) through project grant 1159752 to B.A.E. K.A.H. is an ARC Future Fellow (grant

FT180100123). M.Z. is supported by an Australian Government Research Training Program Scholarship.

We declare no conflicts of interest.

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
