## [Reviewer comments · Microbiology Spectrum]

Microbiology Spectrum

The impact of omega-3 fatty acids on the evolution of *Acinetobacter baumannii* drug resistance

Maoge Zang, Felise Adams, Karl Hassan, and Bart Eijkelkamp

Corresponding Author(s): Bart Eijkelkamp, Flinders University

Review Timeline:

Submission Date:	September 6, 2021
Editorial Decision:	October 3, 2021
Revision Received:	October 14, 2021
Accepted:	October 18, 2021

Editor: Ayush Kumar

Reviewer(s): Disclosure of reviewer identity is with reference to reviewer comments included in decision letter(s). The following individuals involved in review of your submission have agreed to reveal their identity: Helen I Zgurskaya (Reviewer #2)

Transaction Report:

DOI: <https://doi.org/10.1128/Spectrum.01455-21>

October 3, 2021

Dr. Bart A Eijkelkamp
Flinders University
Sturt Rd
Bedford Park, SA 5042
Australia

Re: Spectrum01455-21 (The impact of omega-3 fatty acids on the evolution of *Acinetobacter baumannii* drug resistance)

Dear Dr. Bart A Eijkelkamp:

Thank you for submitting your manuscript to Microbiology Spectrum. Your manuscript has been reviewed by two experts and some modifications are warranted before the manuscript can be considered for publication. I would you to particularly consider the comments provided by Reviewer 1. When submitting the revised version of your paper, please provide (1) point-by-point responses to the issues raised by the reviewers as file type "Response to Reviewers," not in your cover letter, and (2) a PDF file that indicates the changes from the original submission (by highlighting or underlining the changes) as file type "Marked Up Manuscript - For Review Only". Please use this link to submit your revised manuscript - we strongly recommend that you submit your paper within the next 60 days or reach out to me. Detailed information on submitting your revised paper are below.

Link Not Available

Sincerely,

Ayush

Journals Department
Reviewer comments:

Reviewer #1 (Comments for the Author):

In this manuscript, the authors examine the effects of an omega-3 and omega-6 fatty acid on the development of resistance to erythromycin and cross-resistance to tetracycline. Each fatty acid decreased the development of resistance after exposure to erythromycin. Resistance was obtained by mutations that overexpressed either the AdeABC or AdelJK efflux systems. I have the following comments

1. It is stated that 8 ug/ml of erythromycin was subinhibitory, yet this condition selected for enhanced resistance via mutations. Therefore, there was selective pressure. Was a careful growth rate analysis done to confirm that 8 ug/ml was not inhibitory? If not, this should be done and added as a supplemental figure.
2. For AA and DHA, what was the rationale for using the concentrations chosen in this study?
3. The E0 clones + PUFAs have reduced levels of tetracycline resistance, what accounts for this if they had no growth on erythromycin and presumably had no mutations?
4. Figure 3C should include erythromycin MICs for each strain.

5. In Fig. S2, the E0T0 clone appears to have an Erm MIC above wild-type. What accounts for this?
6. Fig. 4A: Some discussion of why mutants other than AdeR I27S might not be isolated in the presence of PUFAs should be added to the discussion. Also the fact that the majority of adeRS mutants did not alter DHA sensitivity in Fig. 4A is not consistent with lines 311-312.
7. Lines 322-324. Supplementation may not be beneficial if it increases resistance by upregulating AdeIJK

Reviewer #2 (Comments for the Author):

This study evaluates an interplay between antibiotic resistance and exposure to polyunsaturated fatty acids in *Acinetobacter baumannii*. The authors report that exposure to fatty acids significantly reduces frequency of resistance to erythromycin and tetracycline and that resistance is associated with overexpression of AdeIJK or AdeABC efflux pumps. The study is well-designed and thorough. The manuscript is well-written and clear. The conclusions are supported by presented data. The findings will be of interest to antibiotic resistance and bacterial physiology fields.

This reviewer noticed a few mistypes (for example on line 46 "association" should be "associated") but these could be fixed during editing.

Staff Comments:

Preparing Revision Guidelines

Please return the manuscript within 60 days; if you cannot complete the modification within this time period, please contact me. If you do not wish to modify the manuscript and prefer to submit it to another journal, please notify me of your decision immediately so that the manuscript may be formally withdrawn from consideration by Microbiology Spectrum.

Reviewer #1:

In this manuscript, the authors examine the effects of an omega-3 and omega-6 fatty acid on the development of resistance to erythromycin and cross-resistance to tetracycline. Each fatty acid decreased the development of resistance after exposure to erythromycin. Resistance was obtained by mutations that overexpressed either the AdeABC or AdeIJK efflux systems. I have the following comments

Q1. *It is stated that 8 ug/ml of erythromycin was subinhibitory, yet this condition selected for enhanced resistance via mutations. Therefore, there was selective pressure. Was a careful growth rate analysis done to confirm that 8 ug/ml was not inhibitory? If not, this should be done and added as a supplemental figure.*

A. We have performed growth analyses of AB5075_UW to examine the impact of various concentrations of erythromycin. These new data have been presented in Figure S2 (Lines 179 – 180; track-changed version) and illustrate that AB5075_UW cells were able to propagate with minimal perturbation when treated with 8 $\mu\text{g}\cdot\text{ml}^{-1}$ erythromycin. Further, the terminology “subinhibitory concentration” has been omitted in the revised version of the manuscript.

Q2. *For AA and DHA, what was the rationale for using the concentrations chosen in this study?*

A. We agree with the reviewer and have taken the opportunity to include this important information in lines 85 - 87 (track-changed version). The concentration of DHA was determined based on typical human plasma DHA levels in individuals on standard Western diet, versus those with higher fish consumption, thereby rendering the concentrations in our study physiologically relevant. The concentration of AA was adjusted for experimental consistency.

Q3. *The E0 clones + PUFAs have reduced levels of tetracycline resistance, what accounts for this if they had no growth on erythromycin and presumably had no mutations?*

A. This is an interesting observation made by the reviewer. We believe that mutations that lead to multidrug efflux pump overexpression have a relatively greater potential on tetracycline resistance as compared to erythromycin, since AB5075_UW is already resistant to erythromycin but remains relatively susceptible to tetracycline. This is reflected in Figure S3, where the gain of tetracycline resistance for the E2T2 clones were approximately 3-fold versus <2-fold for erythromycin. We anticipate that the E0T1 and E0T2 clones harbor mutations possibly associated with increased expression of the *adeB* or *adeJ* efflux systems, sufficient to increase tetracycline resistance, but not erythromycin. Hence, the impact of PUFAs on these groups may be of similar nature and magnitude compared to the E2T2 clones.

Q4. *Figure 3C should include erythromycin MICs for each strain.*

A. We thank the reviewer for this suggestion and have incorporated the data as Figure S3C, where the average MIC values of a larger number of E2T2 mutants has been presented, thereby allowing for appropriate comparison.

Q5. *In Fig. S2, the E0T0 clone appears to have an Erm MIC above wild-type. What accounts for this?*

A. Rather than presenting the modus, we have amended Table 1 to reflect the range of the MIC for erythromycin in strain AB5075_UW. This range is similar across the 22 E0T0 clones (Figure S3).

Q6. *Fig. 4A: Some discussion of why mutants other than AdeR I27S might not be isolated in the presence of PUFAs should be added to the discussion. Also the fact that the majority of adeRS mutants did not alter DHA sensitivity in Fig. 4A is not consistent with lines 311-312.*

A. We agree with the reviewer's comments and have expanded our discussion in lines 318 – 331 (track-changed version). The lack of susceptibility towards DHA in other *adeRS* mutants, but their potential reduced incidence following PUFA co-treatment may be due to the direct impact of DHA upon the transporter, as reported in our previous studies (Zang *et al.* 2021 *mBio*).

Q7. *Lines 322-324. Supplementation may not be beneficial if it increases resistance by upregulating AdeIJK*

A. This is an excellent comment. We have included the following statement "Further, this highlights the potential risk of introducing resistance with PUFA treatment via *adeIJK* upregulation, and would require adequate selection of antimicrobial compounds that may bypass this particular efflux system." in lines 304 - 306 (track-changed version).

Reviewer #2

This study evaluates an interplay between antibiotic resistance and exposure to polyunsaturated fatty acids in Acinetobacter baumannii. The authors report that exposure to fatty acids significantly reduces frequency of resistance to erythromycin and tetracycline and that resistance is associated with overexpression of AdeIJK or AdeABC efflux pumps. The study is well-designed and thorough. The manuscript is well-written and clear. The conclusions are supported by presented data. The findings will be of interest to antibiotic resistance and bacterial physiology fields. This reviewer noticed a few mistypes (for example on line 46 "association" should be "associated") but these could be fixed during editing.

We thank the reviewer for the positive comments and have corrected the typographic errors in the revised manuscript.

October 18, 2021

Dr. Bart A Eijkelkamp
Flinders University
Sturt Rd
Bedford Park, SA 5042
Australia

Re: Spectrum01455-21R1 (The impact of omega-3 fatty acids on the evolution of *Acinetobacter baumannii* drug resistance)

Dear Bart:

Your manuscript has been accepted, and I am forwarding it to the ASM Journals Department for publication. You will be notified when your proofs are ready to be viewed.

Sincerely,

Ayush Kumar
Editor, Microbiology Spectrum
